# Unsupervised Deformable Image Registration with Fully Connected Generative Neural Network

**Ameneh Sheikhjafari**
Department of Computing Science
University of Alberta
Servier Virtual Cardiac Centre
Mazankowski Alberta Heart Institute
sheikhja@ualberta.ca

**Michelle Noga**
Radiology and Diagnostic Imaging
University of Alberta
Servier Virtual Cardiac Centre
Mazankowski Alberta Heart Institute
mnoga@ualberta.ca

**Kumaradevan Punithakumar**
Radiology and Diagnostic Imaging
University of Alberta
Servier Virtual Cardiac Centre
Mazankowski Alberta Heart Institute
punithak@ualberta.ca

**Nilanjan Ray**
Department of Computing Science
University of Alberta
nray1@ualberta.ca

## Abstract

In this paper, a new deformable image registration method based on a fully connected neural network is proposed. Even though a deformation field related to the point correspondence between fixed and moving images are high-dimensional in nature, we assume that these deformation fields form a low dimensional manifold in many real world applications. Thus, in our method, a neural network generates an embedding of the deformation field from a low dimensional vector. This low-dimensional manifold formulation avoids the intractability associated with the high dimensional search space that most other methods face during image registration. As a result, while most methods rely on explicit and handcrafted regularization of the deformation fields, our algorithm relies on implicitly regularizing the network parameters. The proposed method generates deformation fields from latent low dimensional space by minimizing a dissimilarity metric between a fixed image and a warped moving image. Our method removes the need for a large dataset to optimize the proposed network. The proposed method is quantitatively evaluated using images from the MICCAI ACDC challenge. The results demonstrate that the proposed method improves performance in comparison with a moving mesh registration algorithm, and also it correlates well with independent manual segmentations by an expert.

## 1 Introduction

Medical image registration is essential for many clinical image processing tasks Sotiras et al. [2013]. The aim of image registration is to compute a mapping between fixed and moving images by minimizing an objective function that is based on a dissimilarity metric. Recently, promising methods using deep learning have been proposed to improve medical image registration de Vos et al. [2017], Krebs et al. [2017], Li and Fan [2017], Liao et al. [2017]. Deep learning methods such as convolutional stacked auto-encoders have been used to extract features from a pair of images Wang et al. [2017]. Some techniques used supervised deep learning to build prediction model of transformation matrix in order to obtain learning framework Krebs et al. [2017], Rohé et al. [2017], Sokooti et al. [2017],

1st Conference on Medical Imaging with Deep Learning (MIDL 2018), Amsterdam, The Netherlands.

Yang et al. [2017]. These studies use a convolutional neural network regressor to predict the spatial relation between fixed and moving images. Inspired by the recent works in reinforcement learning, methods proposed in Krebs et al. [2017], Liao et al. [2017] applied an artificial agent which uses its own experience to learn and does not require explicitly designed similarity measures. They suggested a reformulation of image registration problem to optimize the deformation model parameters. The agent is trained in a supervised way and explores the space of deformations by choosing an action from a set of actions that update deformation model's parameters.

de Vos et al. [2017] proposed a deformable image registration (DIRNet) using CNNs. This method uses spatial transformer network (STN) Jaderberg et al. [2015] in order to generate a dense displacement field by using local deformation parameters that are estimated by a convNet. DIRNet estimates 2D control points and uses cubic B-splines to represent spatial transformations which register 2D images. Yoo et al. [2017] used bilinear interpolation instead of B-splines to obtain dense spatial transformations and estimated coarse-grained deformation fields at a low spatial resolution. The method can register 2D images by optimizing an image similarity metric derived from an auto-encoder between fixed and transformed moving images. However, predicted coarse-grained spatial transformation which is obtained by these methods might fail to characterize small deformations Li and Fan [2017].

Inspired by fully convolutional networks (FCNs) Long et al. [2015] that facilitate voxel-to-voxel learning, Li and Fan [2017] has proposed a multi-resolution deformable image registration framework based on a deep self-supervised fully convolutional network. This method trains FCNs to estimate voxel-to-voxel spatial transformations to register images by maximizing image-wise similarity metric.

Training data is an important part of deep learning network-based image registration algorithms. There are a variety of strategies that have been proposed to build training data which include simulating synthetic deformation fields and applying them to a set of images to generate new images with known spatial transformations Sokooti et al. [2017]. However, these synthetic deformation fields may not be able to realistically capture the spatial correspondences between actual imagesLi and Fan [2017]. Krebs et al. [2017] proposed a ground truth generator that produces millions of synthetically deformed training samples based on a few real deformation estimations. However, the need for large datasets in the medical field with manual segmentation labels or known spatial transformations is still a major problem in deep learning based algorithms. Thus, we suggest a deep unsupervised network which is able to generate deformation fields independent of training data.

Unlike previous methods that take pairs of fixed and moving images as an input, in our network, inputs are latent, low dimensional vectors. In addition, our method is not trained with known registration transformations, but generates deformation fields and register images by direct optimization of image similarity metric between the fixed and deformed moving image. Furthermore, instead of representing spatial transformations by using B-splines, our method uses bilinear interpolation to capture dense spatial transformations. To the best of the authors' knowledge, this is the first deep learning-based manifold embedding method for unsupervised deformable image registration. Thus, the main contributions of this work are: (1) proposing an unsupervised network and using low dimensional vectors as input (2) generating spatial transformation fields by embedding.

In our formulation, explicit regularization of the deformation field is not used. In our case, the network architecture, the dimension of the latent vector and the $L_2$ norm of the network parameters implicitly regularize the image registration problem. Furthermore, our method can be applied to a sequence of images.

## 2 Method

The objective of the image registration task is to find a spatial transformation that aligns an image pair: a fixed image $I$ and a moving image $J$. It can be formulated as an optimization problem to maximize a similarity metric (or minimize a dissimilarity metric) between the fixed image and the transformed moving image. In a deformable image registration task, the spatial transformation is characterized by a dense deformation field $f$ (or, $\{dx, dy\}$ in 2D) that encodes the displacement vector.

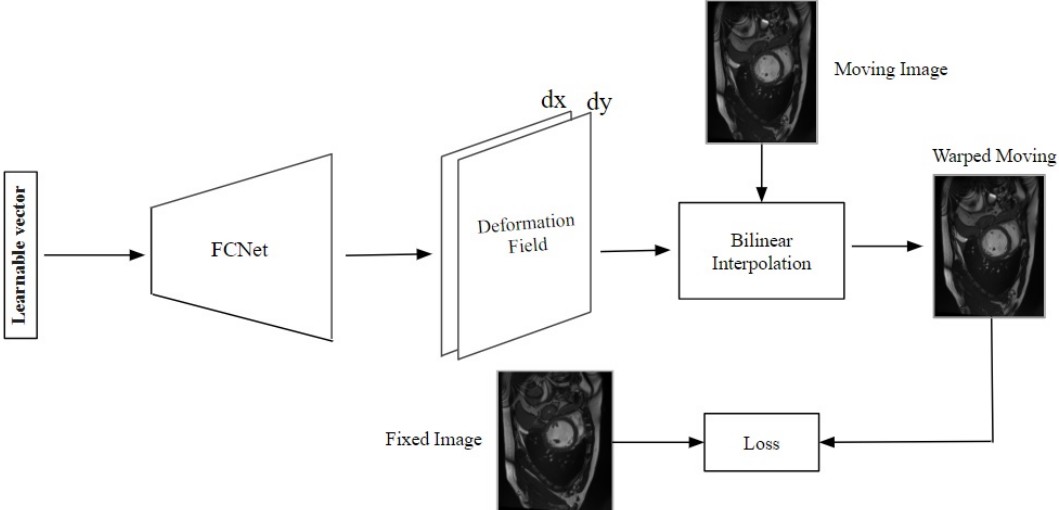

Figure 1: Schematic of the proposed image registration approach based on a fully connected network. The network uses one or more learnable vector which is initialized by a random vector as an input. The fully connected part generates 2-dimensional displacement vector field to warp a moving image to a fixed image. The network is unsupervised and optimizes the similarity metric using backpropagation.

## 2.1 Using fully connected network for optimizing an image dissimilarity metric

We propose a deep network model using FCNet (fully connected network) to solve the optimization problem for image registration. As we use FCNet as a generator, we can directly obtain deformation fields at the same resolution of the images to be registered. In this method, in order to register each image in a sequence, a low dimensional vector (also known as latent vector) is defined as a input vector. One technique to find the latent vectors is using Auto-encoder (AE). The auto-encoder is made of two parts, encoder and decoder. The encoder brings the data from a high dimensional input to low dimensional output. Since using an auto-encoder network to find latent vector add time and computational cost to our framework, we initialized input vectors by random vectors. Then the FCNet applies 8 fully connected layers to generate a two dimensional deformation field $\{dx, dy\}$. The number of kernels (i.e., coefficient matrices) per layer can have an arbitrary size but the number of kernels of the output layer is determined by the dimensionality of the images (e.g. 2 kernels for 2D images that require 2D displacements). FCNet generates a displacement field by minimizing a dissimilarity metric between fixed and warping images using stochastic gradient descent method Adam Kingma and Ba [2014]. The Relu function is used as activation functions throughout. The metric employed here is mean squared intensity difference between a fixed image and a deformed moving image for mono modality image registration. During the optimization, not only the network's parameters, but also input latent vectors are updated. The dense displacement vector fields generated by FCNet are employed to deform a moving image toward the fixed image. We applied bilinear interpolation which has local supports instead of cubic B-spline de Vos et al. [2017] or a thin-plate spline which has a global support. The framework of our method is illustrated in Figure 1.

Suppose $\{I_i\}_{i=1}^n$ is a sequence of images that we would like to register. We have a neural network with parameters $\theta$ that computes a deformation field $f_\theta(t_i) : R^d \rightarrow R^{N \times N}$, where $N$ is the number of pixels and $d$ is a number much smaller than $N$ (in our work $d$ is 25). We can call $f_\theta$ as an embedding function. Thus, for the $i^{th}$ image in the sequence, the neural network takes in a d-dimensional vector $t_i$ and outputs a deformation field $f_\theta(t_i)$. We can warp a moving image $I_i$ by this deformation field to get the warped image $I(f_\theta(t_i))$. Therefore, we can minimize the following cost function for registering image sequence $\{I_i\}_{i=1}^n$:

$$E_{data}(\theta, \{t_i\}_{i=1}^n) = \sum_i |I_i - I_{mov(i)}(f_\theta(t_i))| \tag{1}$$

The minimization is performed jointly over the parameters $\theta$ of the neural network and latent vectors $\{t_i\}_{i=1}^n$, where $I_i$ is a fixed image and corresponding moving image is $I_{mov(i)}$.

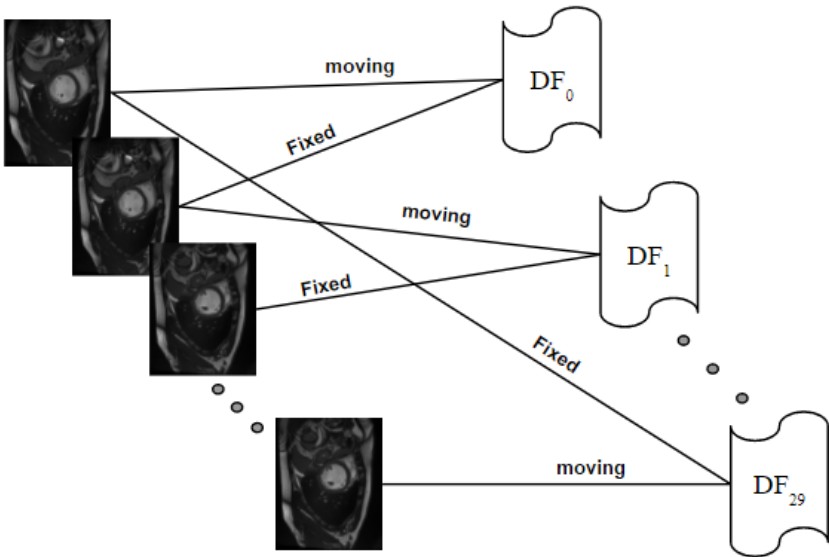

Figure 2: Selection of fixed and moving images in one sequence of the medical image. $DF_i(i = 0, 1, 2, ..., K)$ are deformation fields generated by the network for each pair of fixed and moving images.

## 2.2 Regularization

Image registration is inherently ill-posed so that the existence and the uniqueness of the solution is not guaranteed Myronenko [2010]. Thus, regularization is essential to avoid both physically implausible displacement fields and local minimum during optimization Vishnevskiy et al. [2017]. As the most optimization methods based regularization image registration are typically computationally expensive and time-consuming, an alternative is to regularize network parameters $\theta$. This regularization considers the mean of the sum of squares of the network weights ($MSW$):

$$MSW(\theta) = \frac{1}{N_w} \sum_{n=1}^{N_w} w_n^2, \tag{2}$$

where $N_w$ represents the number of network weight parameters and $w_n$ is an element of the parameter matrix in a vector expression $W$. The weights and biases of the network are initialized as random variables drawn from a Gaussian distribution. Finally, our optimization problem can be formulated as follows:

$$E(\theta, \{t_i\}_{i=1}^n) = E_{data}(\theta, \{t_i\}_{i=1}^n) + \lambda MSW(\theta), \tag{3}$$

The above optimization problem can be solved by backpropagation with stochastic gradient descent.

Our model is implemented using tensorflow. Adam optimization technique Kingma and Ba [2014] is used with learning rate $1 \times 10^{-4}$, image batch size 10 and $\lambda = 0.1$. The results were obtained with NVIDIA GTX 1080 Ti GPU, and 2000 iterations were adopted for the optimization that takes 4–6 minutes per image.

## 3  Experimental Evaluation and Comparisons

We applied our method to 100 short axis cardiac cine MR sequences of 10 patients (30000 images). We compared the accuracy of the proposed method with a moving mesh correspondence algorithm presented in Punithakumar et al. [2017, 2013] by using the same datasets.

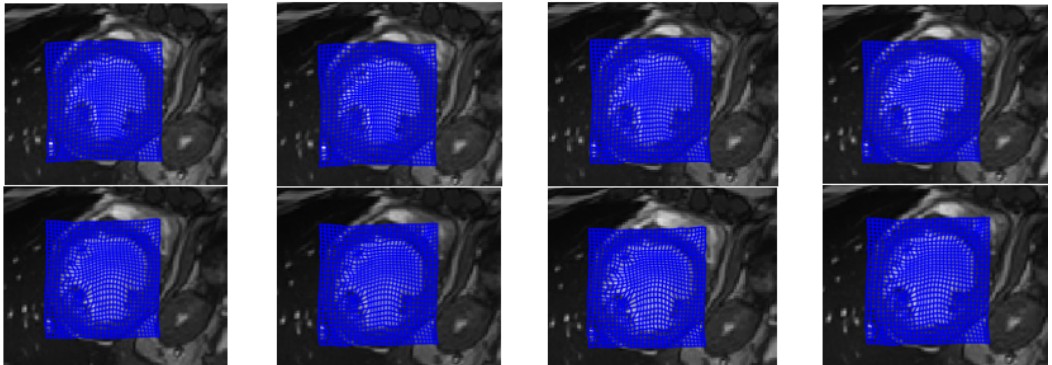

Figure 3: Representative examples of the displacement field obtained by the proposed method

### 3.1 Data

The proposed method is evaluated with clinical Magnetic Resonance Imaging (MRI) images from MICCAI Automated Cardiac Diagnosis Challenge (ACDC) dataset ACDC [2017]. The ACDC 2017 dataset contains 100 exams (all from different patients) that each exam has a sequence with 25–30 frames. Cine MR images were acquired in breath hold with a retrospective or prospective gating and with a steady-state free precession (SSFP) sequence in short axis orientation. A series of short axis slices cover the Left Ventricle (LV) from the base to the apex, with a thickness of 5 to 8 mm and with an interslice gap of 5 mm. The spatial resolution goes from 0.83 to 1.75 mm/pixel. For more details on the dataset, please refer to the ACDC website ACDC [2017].

### 3.2 Evaluation measures

#### 3.2.1 Dice metric

For the geometrical metrics, we use Dice Metric (DM) which usually is used to measure the similarity (overlap) between two surfaces. DM is defined as the ratio of the intersection by the sum of the two regions:

$$D(M, N) = \frac{2(M \bigcap N)}{M + N} \tag{4}$$

where $M$ is the area or volume enclosed by the automatic counters and $N$ is the manual counter. DM varies from 0 (total mismatch) to 1 (perfect match).

#### 3.2.2 Reliability

We evaluate the reliability of our method by using a reliability function (the complementary cumulative distribution function (ccdf)) of the obtained Dice metrics Ayed et al. [2009]:

$$R(d) = P(DM > d), \tag{5}$$

where for each $d \in [0, 1]$, $R(d)$ is ratio of number of images segmented with DM higher than $d$ and total number of images. $R(d)$ measures how reliable the algorithm is in yielding accuracy $d$.

### 3.3 Quantitative Evaluation

The proposed method is evaluated on 100 short axis cardiac cine MR sequences, a total of 30000 images. In each sequence $I_i$ is selected as a moving image and $I_{i+1}$ is selected as a fixed image Figure 2. Our suggestion approach yielded a DM equal to $0.89 \pm 0.03$ for all the data analyzed (DM is expressed as mean $\pm$ standard deviation). Table 1 shows DM statistics for the proposed method and the moving mesh correspondence method Punithakumar et al. [2017]. Using the same data, the method in Punithakumar et al. [2017] yielded a DM equal to $0.85 \pm 0.03$. Figure 6 (a) depicts the DM for a representative sample of the analyzed images. Table 1 reports the accuracies for FCNet in comparison with moving mesh correspondence method and it shows that

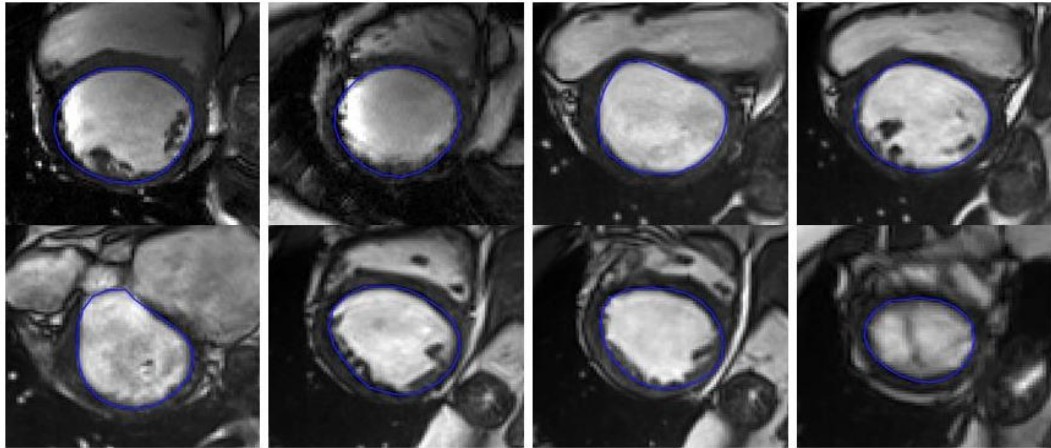

Figure 4: Sample of the boundary results with the proposed FCNet.

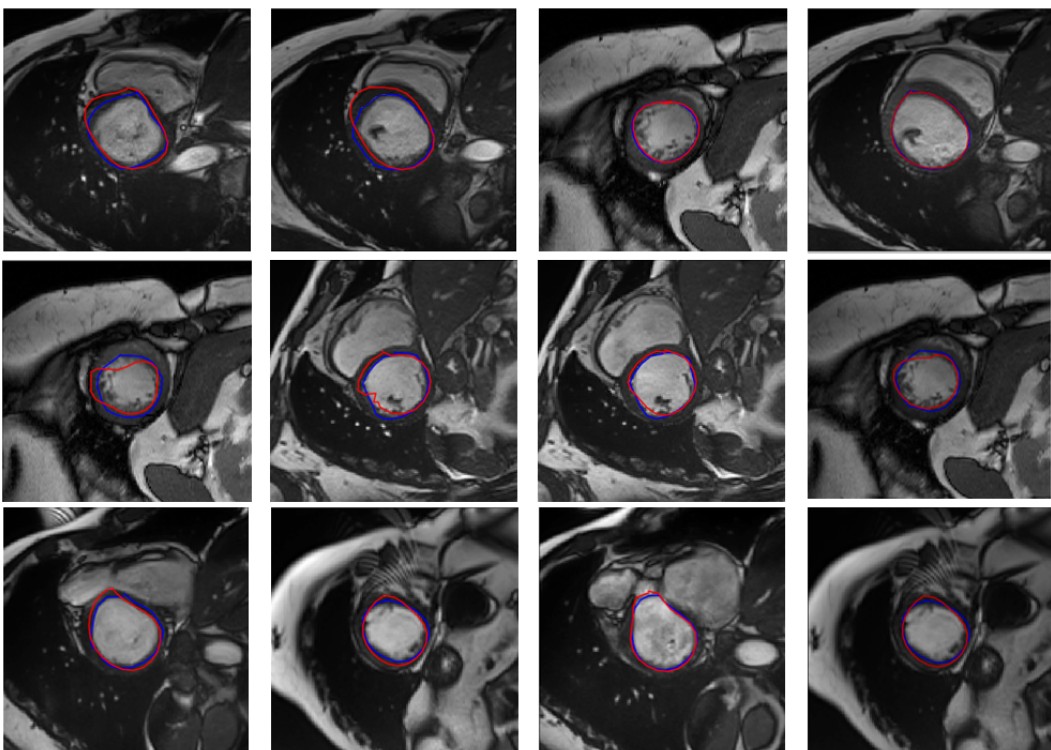

Figure 5: Representative examples of obtained borders of the LV with FCNet (blue) and moving mesh correspondence Punithakumar et al. [2017] (red) methods where FCNet provided significantly more accurate results than Punithakumar et al. [2017].

our approach led to a significant improvement in average the accuracy. In addition, Table 1 reports the reliability of the proposed method and Punithakumar et al. [2017] in different accuracy levels $d = 0.80, d = 0.85, d = 0.90$ and plot $R(d)$ as a function of $d$ in Figure 6 (b). Our algorithm led to a higher reliability curve and improvement in reliabilities.

### 3.4 Visual Assessment

By using a grid mesh, we show the displacement fields obtained by FCNet Figure 3. The comparison of the result of FCNet and moving mesh correspondence method Punithakumar et al. [2017] are given

Table 1: The mean, standard deviation of Dice score and Reliability function ($R(d) = P(DM > d)$). The higher the $DM$ and $R$, the better the performance.

|  | Dice | $R(0.80)$ | $R(0.85)$ | $R(0.90)$ |
|---|---|---|---|---|
| Punithakumar et al. [2017] | $0.85 \pm 0.03$ | 0.95 | 0.66 | 0.06 |
| our method | $0.89 \pm 0.03$ | 1 | 0.91 | 0.44 |

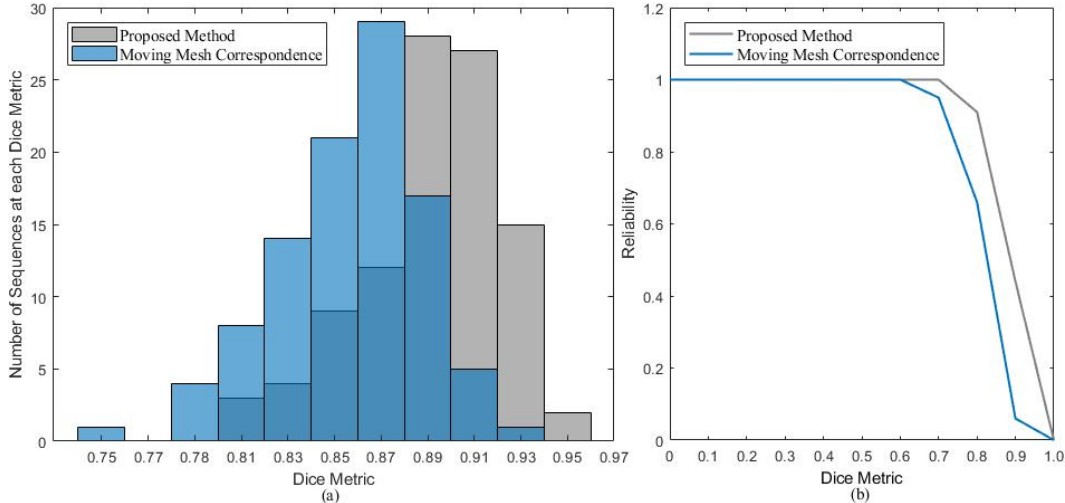

Figure 6: Comparison between proposed method and the methods by Punithakumar et al. [2017] for 100 sequences (30000 images). (a) Dice metric (DM) for the proposed method and Punithakumar et al. [2017]. (b) Reliability $R(d) = Pr(DM > d))$ for the proposed method and Punithakumar et al. [2017]. The proposed method led to a higher reliability curve.

in Figure 5. In Figure 3, we give a representative sample of borders obtained by our method. The FCNet approach yielded more accurate results than the moving mesh correspondence method in a number of image sequences.

## 4 Discussion and Conclusion

In this paper, we proposed a deformable image registration algorithm based on the deep fully connected network to generate spatial transformations. Our network predicted deformation field at the same resolution of fixed and moving images. The image registration optimizes spatial transformation with deep supervision network through feedforward and backpropagation computation. The experimental results show that our method obtained promising performance to register a sequence of cardiac MRI images.

Most deep learning based image registration methods learn spatial transformations from training data with known deformation fields Sokooti et al. [2017], Yang et al. [2017], Krebs et al. [2017], Yoo et al. [2017]. On the other hand, other methods estimate spatial transformations by using the pairs of images (fixed and moving images) de Vos et al. [2017], Li and Fan [2017]. Unlike these methods, our algorithm directly generates spatial transformations by maximizing a similarity metric between fixed and deformed images based on low dimensional learnable vector initialized by random vector, independent from moving and fixed image.

In conclusion, the results have demonstrated that unsupervised deep learning models built upon generative fully connected networks can achieve satisfying performance for deformable medical image registration. This study shows that the proposed approach improves the performance over recent state-of-the-art image registration with respect to accuracy.

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
