# OpenReview forum: "Unsupervised deformable image registration with fully connected generative neural network"
_MIDL.amsterdam/2018/Conference — MIDL 2018 Poster_

### Review · AnonReviewer1 · 2018-04-26
**Application of deep learning to registration with the aim of regularization is interesting, but description of the method is currently ambiguous.**

**Rating:** 2
**Confidence:** 3

**Review:**

The authors propose a deformable image registration method for 2D images based on a neural network. The network is not trained in the conventional supervised way, but optimized at test-time by minimizing the mean squared difference between the intensities of the fixed and warped moving image. This also optimizes a latent vector that is used as an input to the network. The loss function that is employed aims to regularize the deformation field. The method is validated on cine MR images from the ACDC dataset, using the Dice coefficient and a reliability score, and compared to a previous registration algorithm by one of the authors.

The application of deep learning methods to medical image registration is certainly interesting, and the application to regularize deformation fields is a new contribution.

The main concern I have about this paper is that the description of the method is ambiguous. The authors describe that the network and a latent vector are optimized. Since both the vector and the images are inputs to the network it is currently unclear how they relate to each other, and what the purpose of the learnable vector is once it is optimized. It is also unclear why L2-regularization of a network's weights (Equations 3 and 4) will lead to more well-posed deformation fields.

In the introduction, the authors compare their method to other neural network-based methods in literature, stating that the advantage of their method is that no training data is required. However, most conventional image registration methods (i.e. those not based on deep learning) are also unsupervised, and also provide methods for regularization. Are there any other advantages compared to these methods?

Overall, things that would strengthen the paper are a clear explanation of the method and its advantages and disadvantages compared to other registration methods, including those not based on deep learning.

Minor remarks:
- The network architecture is not fully defined in the paper, which makes it impossible to repeat the experiment.
- What is meant by 'Learnable Bilinear Interpolation' in Figure 1?
- The caption below Figure 1 mentions more than one learnable vector can be used, but it is not described to what purpose or how that works.
- The plot in Figure 6a can be made more clear as a boxplot or histogram showing the distribution of the Dice scores. In the current plot it is difficult to discern the individual bars.
- Given that at any given time there is only one pair of images for optimization, how do the authors obtain a batch size of 10 samples?
- The authors mention that the method can be applied to a sequence of images, but it is not explained how this would work.

**Special Issue:**

No

---

### Review · AnonReviewer2 · 2018-04-27
**Review of the Unsupervised Deformable Image Registration using a Generative Model**

**Rating:** 3
**Confidence:** 2

**Review:**

This paper proposes a novel unsupervised deformable image registration using a generative model.  The main contribution of this work is that the deformation fields are generated from a low-dimensional feature representation.  The registration process involves an optimisation that takes this generated low-dimensional feature representation to produce a deformation map, which is then used to optimise an image similarity metric between the fixed and the warped image.  This optimisation estimates both the network parameters and the low-dimensional feature representation that is used to produce the deformation map.  Results on 100 short axis cardiac cine MR sequences of 10 patients (30000 images) show that the proposed method produces better results than Punithakumar et al. [2017], which relies on a moving mesh correspondence algorithm.

The idea explored in the paper is quite interesting and I think it is novel, as far as my knowledge goes.  However, the paper lacks a bit of clarity - for instance, it is not clear if there is a training stage that somehow estimates an anchor value for \theta.  The optimisation that produces the deformation map also learns the values for \theta and t.  This process is not common in the field, and as a result this should be explained in more detail in the text.  Also, even though the papers by de Vos et al. [2017], and Li and Fan [2017] are conceptually different, I believe this submission should have a comparison with them.  Finally, is there any train/test split in the dataset?

Minor issues:
- Above Eq.1, R^d -> R^2N should be R^d -> R^nxn
- It is not clear from Eq. 1, how t_i is initialised - it seems to be randomly sampled from some undefined distribution, but it could be estimated from I_i.  A similar question is how is the inference performed?  For example, from a test image I_i, how is its displacement obtained?
- What is the training and testing sets?  In Section 3, the paper mentions that the method was applied to 100 short axis cine MR sequences of 10 patients, but it is not clear which images were used to train and test the proposed method.
Which optimisation is used: (2) or (4)?
- How can I know if the results shown in Figure 3 are relevant?


**Special Issue:**

No

---

> ### Comment · ~Ameneh_sheikhjafari1 · 2018-05-13
> **Responses**
>
> The idea explored in the paper is quite interesting and I think it is novel, as far as my knowledge goes.  However, the paper lacks a bit of clarity - for instance, it is not clear if there is a training stage that somehow estimates an anchor value for \theta.  The optimisation that produces the deformation map also learns the values for \theta and t.  This process is not common in the field, and as a result this should be explained in more detail in the text.  Also, even though the papers by de Vos et al. [2017], and Li and Fan [2017] are conceptually different, I believe this submission should have a comparison with them.  Finally, is there any train/test split in the dataset?
>
> <Response>
> We agree that the description can be improved towards clarity. In our future work and presentation at the workshop, we plan to include comparisons with deep learning methods. Since our method is unsupervised, we did not have any training / test split for the data.
> <End Response>
>
> Minor issues:
>
> - Above Eq.1, R^d -> R^2N should be R^d -> R^nxn
> <response>
> Thank you for the suggestion. We will replace, R^d -> R^2N with R^d -> R^nxn.
> <End Response>
>
> - It is not clear from Eq. 1, how t_i is initialised - it seems to be randomly sampled from some undefined distribution, but it could be estimated from I_i.  A similar question is how is the inference performed?  For example, from a test image I_i, how is its displacement obtained?
>
> <Response>
> Yes, the t_i can be inferred from I_i, as in an autoencoder. This would be an interesting comparison for our future work.
> <End Response>
>
> - What is the training and testing sets?  In Section 3, the paper mentions that the method was applied to 100 short axis cine MR sequences of 10 patients, but it is not clear which images were used to train and test the proposed method.
>
> <Response>
> The network learns the registration task with an image similarity metric. This method performs iterative optimization for each image pair and there are not training and testing phase.
> <End Response>
>
> Which optimisation is used: (2) or (4)?
>
> <Response>
> I think we should remove the equation that we do not use in the proposed method and mention that in the response.
> <End Response>
>
> - How can I know if the results shown in Figure 3 are relevant?
>
> <Response>
> Figure 3 allows for a visual assessment of the cardiac motion and the corresponding deformation grid line tracking. Unfortunately, the current set of images do not show the tracking adequately. We will replace the figures with a set of images at end-systolic and end-diastolic and corresponding grid lines to show the deformation tracking clearly.
> <End Response>

---

### Review · AnonReviewer3 · 2018-05-09
**interesting application of generative models and neural network**

**Rating:** 3
**Confidence:** 2

**Review:**

In this work an unsupervised deformable Image registration method is proposed. An 8-layer fully connected neural network is used to produce the deformation field for moving image, given a random vector of length 25 as input. The criterion for updating the network parameters is mean squared intensity difference between the moving and fixed image.

Pros:

1- The idea is novel in the sense that it is about fitting a generative net on a few data samples instead of training on a large dataset for registration.
(Ulyanov et al, in Deep Image Prior proposed similar idea but did not apply it for registration.)

2- In order to train the network there is no need to feed the image to the net directly, instead low dimensional vectors are used to represent each image.

3- The proposed model can be trained on a few data samples without supervision, just by minimizing the mean squared intensity difference.

Cons:

1- Exact final configuration of the net is not reported (activation functions, number of neurons in middle layers and value of λ the coefficient of regularization term)

2- To have better evaluation the model should be tested on popular datasets and have a comparison with reported results.
Also a comparison can be done on execution time, since the number of iterations is 2000 for each data sequence and it seems that it's not fast.

3- Since the network is fully connected and the resolution of the generated deformation field is the same as image, it has a large number of parameters and consumes considerable GPU memory which limits the generalizability of the model for using on higher resolution images and also 3D images.

4- Because the initial values of the latent vectors are generated randomly there is no guarantee that we'll get the same results by running the model twice on the same data. If the model was trained on a larger set of samples (Instead of fitting or maybe overfitting the model weights on a few images) the results could be more stable and dependable.



**Special Issue:**

No

---

### Comment · ~Ameneh_sheikhjafari1 · 2018-05-08
**a detailed description of advantages of our method**

In general, we provide a mdical image registration method based on deep
learning. The most important difference between our method and other image
registration methods including deep learning based, is our inputs. Based on our
knowledge, all of those methods use image pairs (fixed and moving images) as an
input and process these images to find deformation field but our model is
generative. An input latent vector generates a deformation field, which is used to
warp a moving image. Our method use low dimensional vectors and generates
deformation filed. For each image pair, we generate a random low dimensional
vector as an input of Fully Connected network. The FCNet generates a
deformation field at the same resolution of the images to be registered. Further,
in our case, the deformations lie on a manifold. The dimensionality of the
manifold and hence regularization for registration can be controlled by
regularizing \theta and the architecture of the network, including the length of
latent vector. This is the reason that we do not require any explicit regularization
of the deformation field.
The main difference between a conventional registration method and our method
is as follows. For a conventional method to register a sequence of images, the
deformation fields do not share any common parameters among them. In our
case, they all share the network parameters \theta. In
addition, not only both de Vos et al. [2017], and Li and Fan [2017] process image
pairs to find deformation field but also, de Vos et al. [2017] uses STN to warp
moving image to the fixed image. Since that, we generate a deformation field at
same resolution of the images to be registered.
In final version, we will explain details of how the method works and difference of
our method to other registration methods.

---

### Decision · Program_Chairs · 2018-05-15
**Paper99 Acceptance Decision**

Poster